# Enhancement of the Fragility Capacity of RC Frames Using FRPs with Different Configurations at Joints

**DOI:** 10.3390/polym15030618

**Published:** 2023-01-25

**Authors:** Saeed Jafari, Seyed Saeed Mahini

**Affiliations:** 1School of Computing, Engineering and Built Environment, Glasgow Caledonian University, London E1 6PX, UK; 2National Center for Timber Durability and Design Life, University of the Sunshine Coast, Sippy Downs, QLD 4556, Australia

**Keywords:** fragility capacity, performance, damage, controlling, FRP-retrofitting, joint, collapse, margin ratio, PGA, web-bonding, flange-bonding

## Abstract

This paper reports the results of an investigation into the effectiveness of different lengths of Fiber-Reinforced Polymer (FRP) sheets in retrofitting the joints of Reinforced Concrete (RC) frames to improve the fragility function of ordinary RC frames. Several 8-storey RC buildings were investigated through FE modelling. The accuracy of the FE models was verified using peer research results. Fragility curves of FRP-retrofitting joints of two referenced RC frames were carried out by OpenSees, through Incremental Dynamic Analysis (IDA) analysis under 22 far-field earthquake records from 0.1 g to 4.0 g (with 0.1 g interments), based on FEMA P-695. Two types of retrofitting methods, web and flange bonding, were modeled and studied. The results showed that the fragility capacity of the retrofitted RC frames was significantly improved. Moreover, frames with longer sheets of FRP showed increased performance. In the complete state, the range of probability of exceedance grew from 2–2.5 g to 3–3.5 g (nearly 1 g), whereas, in the minor state, this growth was nearly 0.05 g. However, the fragility function of the flange-bonding was enhanced at a higher rate compared with that of the web-bonding RC frames. Carbon Fiber-Reinforced Polymer (CFRP) and Glass Fiber-Reinforced Polymer (GFRP) materials improved the probability of exceedance of the complete state from 3 g to 4.5 g and 4.8 g in flange bonding frames. This enhancement for both types of frames was more significant when joints were retrofitted with 400 and 500 mm compared with 600, 700, and 800 mm. The midpoint of the PGA at the complete damage state in the web-bonding frame increased from 1.076 g to 1.664 g and in the flange-bonding frame retrofitted with GFRP and CFRP raised from 1.551 g to 2.769 and 3.076, respectively. The collapse margin ratio (CMR) indicates an acceptable improvement in the retrofitted frames. Overall, the rate of enhancement in fragility function from the original frame to the frame with 500 mm FRP was significant; however, the slope of this rate declined for longer FRP sheets. The fragility performance improvement resulted in controlling plastic hinging by FRPs.

## 1. Introduction

The enhancement of the seismic behaviour of existing RC structures through applying FRP sheets is a popular method in which a composite concrete composite section is produced. In many cases, rehabilitation of some buildings which have been designed for gravity loads and are based on older seismic design codes is necessary. The main reasons for the popularity of the mechanical properties of FRP materials are their lightweight nature, corrosion resistance, strength, stiffness, and applicability to concrete surfaces. CFRP and GFRP have varied benefits and key applications as a result of their disparate composition, performance, and cost. For example, FRPs based on CFRP have a significantly higher strength, excellent fatigue resistance, corrosion resistance and creep resistance, and are also lighter due to their lower density. However, a significant disadvantage of carbon fiber is its high cost and limited elongation at break. Alternatively, GFRP is cheaper and has increased elongation, which is a reason that GFRP is often used, although it performs worse than CFRP, and also suffers from poor corrosion resistance [1,2].

FRP materials can be used on beams, columns, connections, and shear walls, which leads to improved lateral load carrying capacity and reduced drift and/or increased ductility [3]. Nowadays, various methods of evaluating the seismic vulnerability of structures have been developed by researchers [4]. Fragility curve analysis has been widely used as a probabilistic indicator of structural safety against earthquakes [5]. This parameter determines the performance of a structure subjected to earthquake load at different damage states [6]. A referenced two-dimensional, eight-story, three-bay, RC moment-resisting frame, which was retrofitted by Maheri and Akbari [7] using a steel-bracing system, was retrofitted at joints with Web-bonded CFRP sheets and the results of nonlinear pushover analysis were compared with those of the original frame and the steel-braced frame. Nonlinear pushover analysis is a well-known method to represent the seismic performance of the structures. In a numerical study, Zou et al. [8] presented an optimisation technique for the performance-based design of seismic FRP-retrofitted RC building frames. Their numerical studies showed that the seismic resistance of a RC frame designed for gravity loads only can be significantly enhanced over confinement of columns using FRP jacketing. They indicated that FRP confinement increases the strength of columns but has little effect on their stiffness. This is an important advantage in seismic retrofitting, as larger stiffnesses lead to higher seismic forces.

Di Ludovico et al. [9] carried out pushover analysis of the FRP-retrofitted frame tested under bi-directional seismic loading and validated the results with experimental data, confirming the effectiveness of the nonlinear pushover analysis. Niroomandi et al. [3] studied the effects of Web-bonded FRP-retrofitting of the joints on improving the structural ductility of ordinary RC frames. The retrofitted joints’ stiffness in the form of the moment-rotation relation was first determined by a detailed finite element (FE) modelling of the composite joints and the results were then utilised to carry out nonlinear pushover analysis of the full frame to evaluate its force–displacement capacity curve. The capacity curve was then utilised to evaluate the seismic performance parameters of the frame. Niroomandi et al. [3] reported that retrofitting with 500 mm FRP sheets can enhance the seismic performance level and the seismic behaviour factor (R) of ordinary RC frames. Obaidat et al. [10] investigated the behaviour of repaired RC beam-column joints using CFRP plates with a variety of retrofitting schemes. Their results showed that all repaired joints improved the strength of the original joints and even enhanced their strength capacity. Pohoryles et al. [11], in a state-of-the-art review, revealed several promising features of FRP-strengthening schemes for beam-column joints. They highlighted the important impact of realistic size, loading, and geometry of the testing specimens on the effectiveness of FRP retrofitting. It was shown that, on average, the effectiveness of repairing pre-damaged joints was similar to that of retrofitting specimens without damage.

Various methods, such as fragility curve analysis, have been developed by researchers for evaluating the seismic vulnerability of structures [2]. Abdelnaby [12] evaluated the seismic fragility relationship of an RC frame subjected to mainshock–aftershock sequences. Their results showed that the impact of multiple earthquakes can increase the vulnerability of RC buildings significantly. Structures with irregularity in the plan were also studied by Moon et al. [4]. They applied a different method to derive the fragility curve of a three-dimensional model. Moreover, the seismic collapse fragility analysis of RC frames with shear walls was studied by Dabaghi et al. [13]. They performed IDA analysis based on FEMA P695 for far-field records. Xian et al. [14] evaluated the seismic collapse safety and retrofitting low-ductility of RC frames. These researchers compared two retrofitting schemes, including steel-brace retrofitting and increasing the column section. Steel-brace retrofitting showed an increased improvement in seismic response to earthquakes. Heshmati et al. [15] studied the effect of far-field and near-field earthquake records on the seismic design of IMRF and SMRF structures according to FEMA P695 methodology. They showed that the response modification factors introduced in seismic the design code for IMRF and SMRF under far-field records met the requirement criteria of FEMA P695. However, in the near-field earthquakes, only SMRF had an acceptable behaviour.

Omranian et al. [16] studied the seismic damage of an RC-skewed bridge subjected to ground motions using fragility curves in the original form and after being retrofitted with FRP. Their results showed that the strengthening method enhanced the fragility of the bridge. Naseri et al. [17] investigated an analytical method for the improvement of the fragility curves of a strengthening RC box-girder bridge. They confirmed that steel jackets and FRP sheets reduced the fragility vulnerability of the bridge. Shafaeu and Naderpour [18] investigated the seismic fragility of an FRP-retrofitted ordinary RC frame under mainshock–aftershock earthquakes. Their results showed that although FRP improves that fragility function of the frame, the reduction capacity due to the earthquake records is independent of the strength and ductility of the structure. Devi et al. [19] compared the fragility curves obtained from a RC structure before and after strengthening by FRP. Their retrofitting strategy increased the capacity of the RC frame. Zhang et al. [20] developed a fragility analysis for performance-based blast design of FRP-retrofitted RC columns. They compared the results of both the original and retrofitted frames and provided a method to predict the failure probabilities of RC columns under blast loads.

In the current study, the effect of different lengths of FRP sheets applied to the joints on the fragility capacity of an RC frame was investigated. The effective length is an important parameter on the fragility behaviour of RC frames, which has received less prior investigation. Overall, various studies show that FRP materials can enhance the seismic performance of buildings. However, different patterns and configurations of FRP materials have different impacts on the behaviour of the frames. Fragility performance is one of the factors that has not been evaluated as much as is needed. Therefore, this study examined the impact of various configuration of FRP at the joints in terms of improving the fragility function of RC frames.

Accordingly, the main objective of the current study was to investigate the effects of strengthening the joints by different lengths of FRP sheets ranging from 400 mm to 800 mm on the seismic performance parameters of ordinary RC frames and the fragility capacity of the retrofitted frames. The retrofit schemes that were considered are those involving CFRP web-bonding and flange-bonding of the frame joints. The web and flange-bonding of the considered beam are similar to the scheme used previously by a number of investigators, including those reported in [21,22,23,24,25,26,27]. The variation of the median of PGA on each damage state as well as the collapse margin ratio (CMR) in the original and retrofitted frames were also studied.

## 2. Selected Frames

### 2.1. Web-Bonding Frame

The complete details of the sections of the studied frame are shown in Figure 1 and Table 1. The moment-resisting frame is shown in Figure 2, previously designed by Maheri et al. [7,28], in which the equivalent static earthquake load was based on the Iranian seismic code [29]. The moment-resisting frame was designed based on the strong column–weak beam principle using ACI-318-95 Code [30] and the steel bracing system was designed using the AISC-LRFD Code [31,32].

To study the plastic hinging formations, the selected frame was modelled with FRP web-bonded joints as shown in Figure 3. The schematic scheme of the web-bonding FRP retrofit configuration for an interior joint is shown in Figure 3.

According to Pauley and Priestly’s design approach, a trial-and-error procedure was adopted to find the best thickness of FRP with 500 mm length to relocate the plastic hinge in beam elements for the above-mentioned frame [3,33]. Table 2 represents the mechanical properties of the CFRP material used in the modelling. The thickness of FRP used in the retrofitted frames is higher for column-beam joints in larger sections. To improve the performance, the thickness of FRP sheets in joints with stronger sections is greater than the thickness of sections of less strength [3]. The thickness of the FRP sheets were as given in Table 3.

### 2.2. Flange-Bonding Frame

The frame considered in this part of the investigation was an 8-storey moderate moment-resisting RC building designed by Ronagh and Eslami’s modelling and analysis in SAP2000 [34]. The seismic loads were based on Iranian seismic code [29] similar to UBC 1994 [35], the peak ground acceleration was assumed to be 0.3 g, and the soil was type III, similar to class D of FEMA-356 [36]; moreover, the dead and live design loads considered were 30 kN/m and 10 kN/m, respectively. In addition, the compressive strength of concrete was taken as 25 MPa and deformed bars of Grade 60 (f_y_ = 420 MPa) were chosen for the steel reinforcement. The designed beam and column sections with reinforcements are presented in Table 4 and Figure 4. The elements’ dimensions and assigned sections are shown in Figure 5.

To study the plastic hinging formations in this frame, a typical flange-bonded joint with CFRP and GFRP is shown in Figure 6. The schematic scheme of the flange-bonding FRP retrofit configuration for an interior joint is also shown in Figure 5.

## 3. Numerical Modelling

### 3.1. OpenSees Modelling

To model FRP sheets and study the results, the RC frames were modelled using OpenSees. The Open System for Earthquake Engineering Simulation (OpenSees) is a software framework for simulating the seismic response of structural and geotechnical systems. OpenSees can model various types of elements as well as their connections built with different types of uniaxial materials and sections under a wide range of loading patterns and linear and nonlinear analyses. Furthermore, OpenSees has advanced capabilities for analysing the nonlinear response of systems through different types of solution algorithms and methods [37]. The definition of the modelling including geometry, materials, sections, etc. was achieved using The Tcl scripting language because of its capabilities [38].

#### 3.1.1. Materials

Concrete, steel, and FRP sheets were modelled through uniaxial material commands. The UniaxialMaterial concrete01 command was used to construct a uniaxial Kent–Scott–Park concrete material object with a degraded linear unloading/reloading stiffness according to the work of Karsan-Jirsa and no tensile strength. Steel bars were modelled by UniaxialMaterial Steel01, which is used is to construct a uniaxial bilinear steel material object with kinematic hardening and optional isotropic hardening described by a non-linear evolution equation. Elastic command was used to model FRP sheets. This command is used to construct an elastic uniaxial material object. Through Minmax material command, the stress–strain behaviour of the FRP material was defined. If, however, the strain ever fell below or above certain threshold values, the other material was assumed to have failed. From that point on, values of 0.0 were returned for the tangent and stress [38].

#### 3.1.2. Elements

Non-linear elements in the original frame were modelled using nonlinear BeamColumn elements. This command is used to construct non-linear beam or column element objects, which are based on non-iterative or iterative force formulation, and which consider the spread of plasticity along with the elements. This command allows the determination of the location and the section of elements. To model the elements in the FRP-retrofitted frame, the Beamwithhinges element command was used. This command was used to construct a forceBeamColumn element object, which is based on the non-iterative (or iterative) flexibility formulation. The locations and weights of the element integration points are based on the ‘so-called’ plastic hinge integration, which allows plastic hinge lengths and sections to be specified at the element ends. To model braces in X-braced frames, truss element was used. PDeltaCrdTransf was used to construct the P-Delta Coordinate Transformation object, which performs a linear geometric transformation of beam stiffness and resisting force from the basic system to the global coordinate system, considering second-order P-Delta effects [38].

#### 3.1.3. Recorders

To generate base shear–roof displacement diagrams by pushover analysis, two recorders with reaction and displacement arguments were used. These recorders can record all reactions and displacements in all degrees of freedom and direction of any node in the model during the analysis. Recorders with plastic rotation arguments were defined to record the rotations of the end of elements. After recording the roof displacement base reactions and plastic deformations for all elements, the base shear–roof displacement and plastic hinge distribution diagrams were drawn using MATLAB [39] and FEMA356 [31,36].

### 3.2. Numerical Modelling Validation

In order to verify models created in OpenSees, nonlinear pushover analysis was carried out for four frames, which were studied by Niroomand et al. [3] considering their assumptions and design codes. Based on a study by Mwafy and Elnashai [40], the inverted triangular distribution of lateral loading was used in the pushover analysis. Force-deformation criteria for plastic hinging were defined according to FEMA356 [36] patterns. The base shear versus roof displacement curves for the original and the CFRP web-bonding frame, as well as the flange-bonding CFRP and GFRP frames modelled via OpenSees, were presented and compared with the results of Niroomandi et al. [3] and Ronagh and Eslami [34] in Figure 7 and Figure 8, respectively. As can be seen from both figures, the results are close, which shows that there is a good agreement between the outcomes of the pushover analysis and also indicates that the OpenSees modelling is reliable. The maximum shear capacity of original and wed-bonded CFRP frames in SAP2000 were 72.3 and 105.8 ton, while they were 77.4 and 98.7 ton in Opensees models, respectively (Figure 7). For flange-bonding frames, a comparison between CFRP and GRFP materials was carried out to study the effect of different materials. As Figure 8 shows, the maximum base shear obtained from the SAP2000 models for the original and flange-bonded GFRP and CFRP frames were 139.7, 200, and 247.9 ton, respectively, whereas from the OpenSees models, these were 118.5, 189.9, and 234 ton, respectively. Some fluctuations were observed between the SAP2000 and OpenSees models, which are related to the fact that there is no direct method to model FRP materials in SAP2000, while this is possible with OpenSees.

## 4. Fragility Analysis

The selected frames were analysed by far-field records based on FEMA P695 [41] to determine the collapse capacity and the collapse-margin ratio (CMR). In the IDA analysis, a series of nonlinear dynamic analyses were performed for each seismic record to predict the complete response range. The recommended number of records for deriving fragility curves is 10 to 20 [42]. Therefore, in this study, 22 individual far-field records were used in FEMA P695 in accordance with Table 5. The IDA analysis was carried out from 0.1 g to 4 g in 0.1 g increments. Accordingly, 880 IDA analyses were performed for each frame (more than 12,000 analyses for all frames). The maximum drift of each frame was obtained from IDA analyses to produce the fragility curves.

Using MATLAB, the fragility curve for all frames for slight, moderate, extensive, and the complete structural damage states were extracted from the outcome of the IDA analyses. The probability of exceeding the above-mentioned states was modeled as a cumulative lognormal distribution as [43]:(1)$$p\left(:\leq D\right)=\Phi \left[\frac{1}{\beta_{sd}}\ln\left(\frac{S_{d}}{S_{c}}\right)\right]$$
where: *P* is the probability of exceeding a damage state (*D*) based on the maximum story drift, Φ is the standard normal cumulative distribution function, $\beta_{sd}$ is the standard deviation of the natural logarithm of spectral displacement of damage state, obtained from Equation (2), $S_{c}$ is the median value of spectral displacement at which the frame reaches the threshold of the damage state presented in Table 6, and $S_{d}$, is the spectral displacement which can be obtained Equation (3) [42,44].
(2)$$\beta_{sd}=\sqrt{\frac{1}{N}\overset{N}{\underset{i}{\sum }}{\left(x_{i}-\mu \right)}^{2}}$$
(3)$$\ln\left(S_{d}\right)=a\ln\left(PGA\right)+b$$
where: *a* and *b* are regression coefficients, obtained from logarithm regression analysis of maximum drift in different PGAs. As the number of stories of all frames is eight, based on HAZUS-MH MR-5 [43], these structures are considered as high-rise types; thus, the structural fragility curve parameters (for moderate code) are in accordance with Table 6.

### 4.1. Fragility Curves of Web-Bonding Frame

The results of the fragility analyses of original and web-bonded frames with 400 to 800 mm of CFRP sheets are presented in Figure 9. To study the effect of the length of FRP sheets in the web-bonding frames, the fragility curve of each state was compared and shown Figure 10. In all states, the fragility function of the original frame improved. Applying 400 mm of CFRP on the web of joints of the frame caused a slight enhancement. However, changing the length of CFRP sheets to 500 mm led to a significant improvement. Using longer FRP sheets increased the seismic capacity at a low slope rate.

To obtain a better understanding regarding the enhancement of the fragility capacity of all the web-bonded frames, the PGAs of the probability of 50% and 100% of occurrence of each state are compared in Table 7. The probability of reaching complete collapse in the original frame is 2.458 g of PGA. However, applying 400, 500, 600, 700, and 800 mm of CFRP on the web of joints enhanced this parameter to 2.831, 3.188, 3.307, and 3.489 g, respectively. Therefore, it can be concluded that the effectiveness of CFRP sheets on the web of the joints is higher at the critical states.

### 4.2. Fragility Curves of Flange-Bonding Frame

The results of the fragility analyses of the original and flange-bonded frames with 400 to 800 mm of GFRP and CFRP sheets are presented in Figure 11 and Figure 12, respectively. Figure 13 and Figure 14 compare the results of fragility analyses in each individual state. The behaviour of frames was similar to the web-bonding one. However, the rate of enhancement was greater. The improvement of the seismic response and fragility capacity occurred in flange-bonding frames. The impact of CFRP materials on the fragility curves of frames was greater when compared with GFRP. Unlike web-bonding frames, a significant improvement was observed even in sheets with 400 mm. The same growth occurred in 500 mm FRP. However, similar to the web-bonding frames, the longer FRP sheets showed a minor enhancement. The impact of applying FRP in the higher seismic levels was greater compared with the minor and moderate states. For the GFRP and CFRP, flange-bonded frame summaries of the fragility curves are presented in Table 8 and Table 9. The required PGA values to reach 50% and 100% of each state for all frames were compared. In the original frames, the probability of 100% complete collapse occurred with 3.069 g of PGA. However, applying 400, 500, 600, 700, and 800 mm of GFRP on the flange of the joints improved this parameter to 3.556, 4.159, and more than 4.500 g of PGA. The enhancement in the FB-CFRP frames was more noticeable, whereas, using 400 mm of CFRP improved the required PGA to 4.350 g and using 500 mm and, moreover, enhanced this parameter up to 4.500 g. The reason for a better enhancement of CFRP materials compared with the GFRP material is its higher modulus of elasticity and tensile strength.

## 5. The Effect of FRP Retrofitting on the Median of PGA in Damage States

The value of the median PGA in minor, moderate, extensive, and complete states for all frames are presented in Figure 15, Figure 16 and Figure 17. For the web-bonding frame, the median PGA in the minor state was 20%, moderate state 43%, extensive state 80%, and complete state 55% enhanced. The median PGA in all states in FRP with 600, 700, and 800 mm was close, whereas, the improvement rate in FRP with 400 and 500 mm was noticeable. Similar behaviour was observed in the flange-bonding frames in which the median PGA in the minor state was 35%, moderate state 49%, extensive state 66%, and complete state 79% enhanced in the GFRP-retrofitted frame. In the CFRP-retrofitted frame, this parameter improved to 66%, 81%, 115%, and 98%, respectively.

## 6. The Effect of FRP Retrofitting on CMR

The collapse margin ratio (CMR) is a comprehensive tool that reflects the complete seismic capacity of buildings [15]. This parameter, based on the FEM P695 report [41], is defined as the ratio of the median intensity of a collapse level ground motion (complete damage state),$\;{\hat{S}}_{CT}$ to the spectral acceleration with the MCE and collapse level earthquake, $S_{MT}$. CMRs are calculated using Equation (4):(4)$$\mathrm{CMR}=\frac{{\hat{S}}_{CT}}{S_{MT}}$$

${\hat{S}}_{CT}$ was obtained from the fragility curve and IDA analysis and $S_{MT}$ was determined based on Iranian seismic code 2800 [29]. The CMRs of all frames in the complete damage state are presented in Figure 18. Using FRP effectively improved this parameter. The rate of improvement in FRP with 400 and 500 mm lengths in all frames experienced a significant increase. Applying longer FRP sheets led to a moderate improvement. The flange-bonded frames showed better performance in comparison with the web-bonding frames. Overall, CMR was improved by 55%, 74%, and 92% in CFRP web-bonding, GFRP, and CFRP flange-bonding frames, respectively.

## 7. The Relationship between Plastic Hinging and Fragility

To show the impacts of the plastic hinges on the fragility capacity of frames retrofitted with different lengths from 400 mm to 800 mm, the plastic hinge distributions and their performance levels for the original and FRP retrofitted frames at the target displacement point are given in Table 10. The performance levels of the joints were obtained graphically from force–displacement capacity curves and the results are shown in Table 10. The letters B to E and notations IO (immediate occupancy), LS (life safety), and CP (collapse prevention) represent these performance states.

In terms of plastic hinging, although the number remained the same, the level of plastic hinging decreased. The state of plastic formation improved from E to D for the frame with 400 mm FRP. This improvement led to improvement in the fragility capacity in extensive and complete damage states, while, in the minor and moderate states the improvement was minimal. Using 500 mm of FRP reduced the number of plastic hinges in D and LS states noticeably, which resulted in a considerable improvement in all damage states from minor to complete. However, the number of plastic hinges in frames in D and LS states decreased by a small rate by applying longer FRP sheets from 600 to 800 mm. This shift has a slight impact on the fragility response of the frame and a minimal improvement was observed.

The number of plastic hinges in the flange-bonding frame with 400 mm of GFRP in beams as well as columns decreased in severe states in comparison with the original frame. Applying longer GFRP sheets resulted in eliminating plastic hinges in beams and columns with level E for 500 mm of GFRP. This shift led to the improvement of the fragility response in all states compared with the original and 400 mm frames. The number of plastic hinges in frames with 600 to 800 mm of GFRP in critical states remained the same; thus, the fragility capacity of these frames showed minimal improvement. Similar behaviour was observed in flange-bonding CFRP-retrofitted frames. However, in the 400 mm frame, controlling the plastic hinge has been more successful, and the number of plastic hinges at the LS level decreased at a higher rate in comparison with the similar frame with GFRP; thus, the improvement of fragility capacity at 400 mm noticeably happens for all damage states. Moreover, using longer CFRP sheets on the flange of connections has a small impact on the number of plastic hinges in severe states with minimal improvements in the level of beams and columns led to a small enhancement of the fragility curve in frames with 500 to 800 mm CFRP.

## 8. Controlling Debonding Effect

Previous studies have shown that debonding failure can occur in FRP sheets before the loads reach the expected values, which leads to unsatisfactory enhancement effects and decreases the load-carrying capacity of the rehabilitated elements [45]. Therefore, it is vital to control the debonding failure of FRP-retrofitted joints particularly at the end of the sheets. In this study, the maximum strain that occurred in FRP sheets obtained from OpenSees was compared to the maximum values recommended in ACI 440.2-2017 [46] and the proposed equations by Chen and Teng [47] and Teng et al. [48]. The effective FRP strain is limited by the following:(5)$$\varepsilon_{fd}=0.41\sqrt{\frac{f_{c}^{’}}{n\cdot E_{f}\cdot t_{f}}}\leq 0.9\varepsilon_{fu}$$
where, *ε_fd_* is the maximum strain recommended and less than that which the FRP sheet is prevented from debonding, *f′_c_* is the 28 day concrete compressive strength, *n* is the number of sheet layers, and *E_f_* and *t_f_* are the elastic modulus and the thickness of the FRP sheet, respectively. Table 11 and Table 12 indicates the maximum effective FRP strains that occurred at the end of the sheets obtained from the models in OpenSees for web-bonded frames and Flange-bonded frames as well as the amounts derived from Equation (5) where these amounts are compared with the limiting values allowed by ACI 440.2-2017. These tables confirm that the probability of occurrence of local debonding failure at the end of the sheets due to shear or normal stress concentrations is highly unlikely.

## 9. Conclusions

This study aimed to investigate the improvement of fragility capacity and function of a referenced eight-storey RC frame FRP retrofitting at joints. Web-bonding and flange-bonding FRP retrofitting with various lengths of FRP sheet were modelled and analysed under 22 far-field earthquake records, based on FEMA P695. Fragility curves, the median of ground acceleration of each state, and the collapse margin ratio of the selected frames were carried out. The conclusions are as follows:Applying FRP sheets at joints improves the fragility capacity of RC ordinary frames. This method of rehabilitation has greater effects in the more severe damage states. The Flange-bonding frames indicated a better performance in comparison with the web-bonding system. Moreover, in general, it can be concluded that FRP material with 500 mm can lead to a significant improvement in the response of frames, whereas FRP materials with 400 mm show a slight improvement. Additionally, the longer FRP material sheets progressively strengthen the frames at a slight rate. Overall, the relationship between FRP sheets length and behaviour of frames is similar to an exponential function.The probability of complete collapse (complete state) in web-bonded frames changes nearly 1 g from 2–2.5 to 3–3.5 g. Additionally, the retrofitted joints improved the median of PGA in the complete state by 55%. In addition, the CMR in the web-bonding frame was enhanced by 55%.In the flange-bonding frame, the GFRP and CFRP materials increased the PGA of the complete state from 3 g to 4.5 and 4.8 g, respectively. The median PGA at this level was improved by 79% and 98%, and 74% and 92% in CMR in the GFRP and CFRP flange-bonded frames was observed, respectively.Improving the fragility performance resulted in controlling plastic hinging by FRPs, particularly in the sever states from LS to E, where the number of plastic hinges declined.

## Figures and Tables

**Figure 1 polymers-15-00618-f001:**
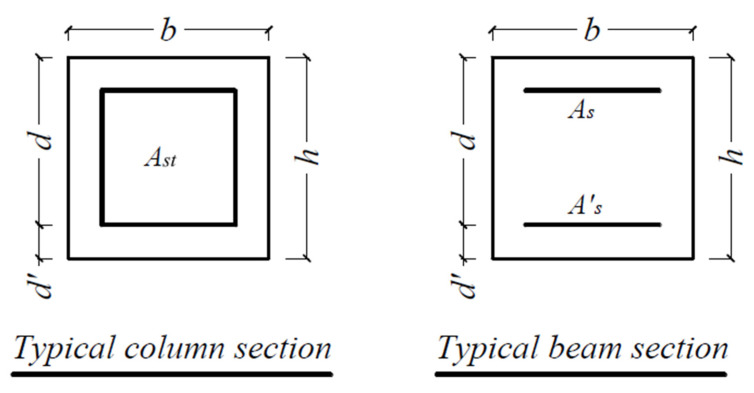
Distribution of longitudinal reinforcement in a typical beam and column section in web-bonding frames [7,28].

**Figure 2 polymers-15-00618-f002:**
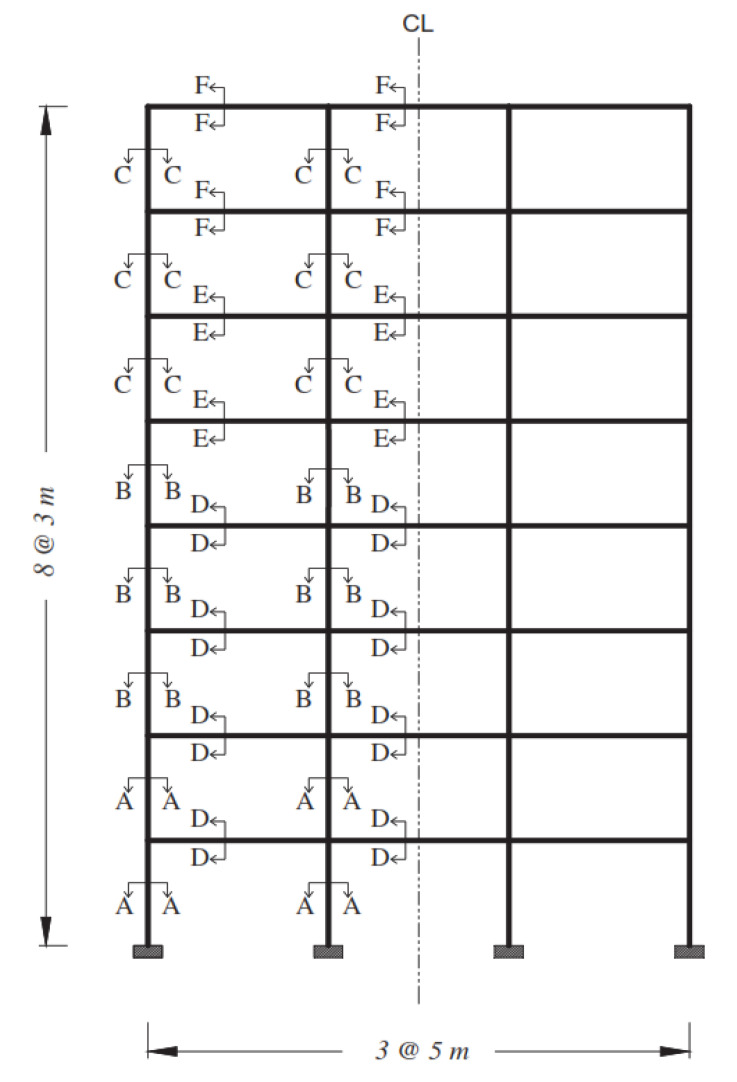
Selected moment-resisting frame [7,28].

**Figure 3 polymers-15-00618-f003:**
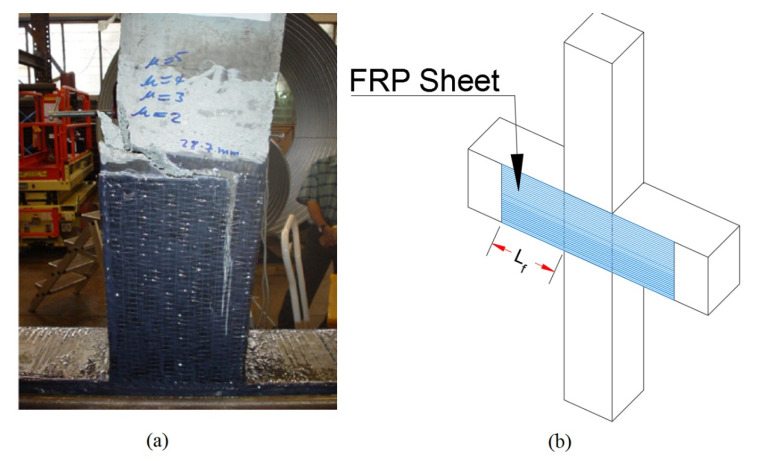
(**a**) Web-bonding CFRP retrofitting system [26,27]. (**b**) Schematic scheme of web-bonding CFRP retrofit in an interior joint.

**Figure 4 polymers-15-00618-f004:**
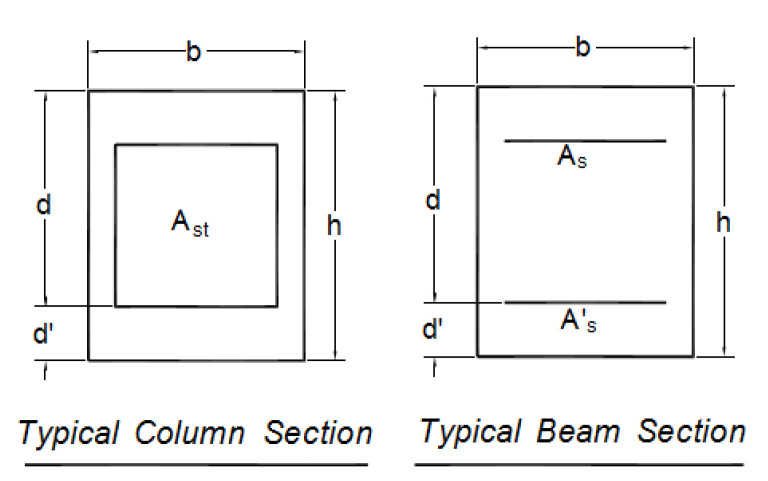
Distribution of longitudinal reinforcement in a typical beam and column section in the flange-bonding frames [34].

**Figure 5 polymers-15-00618-f005:**
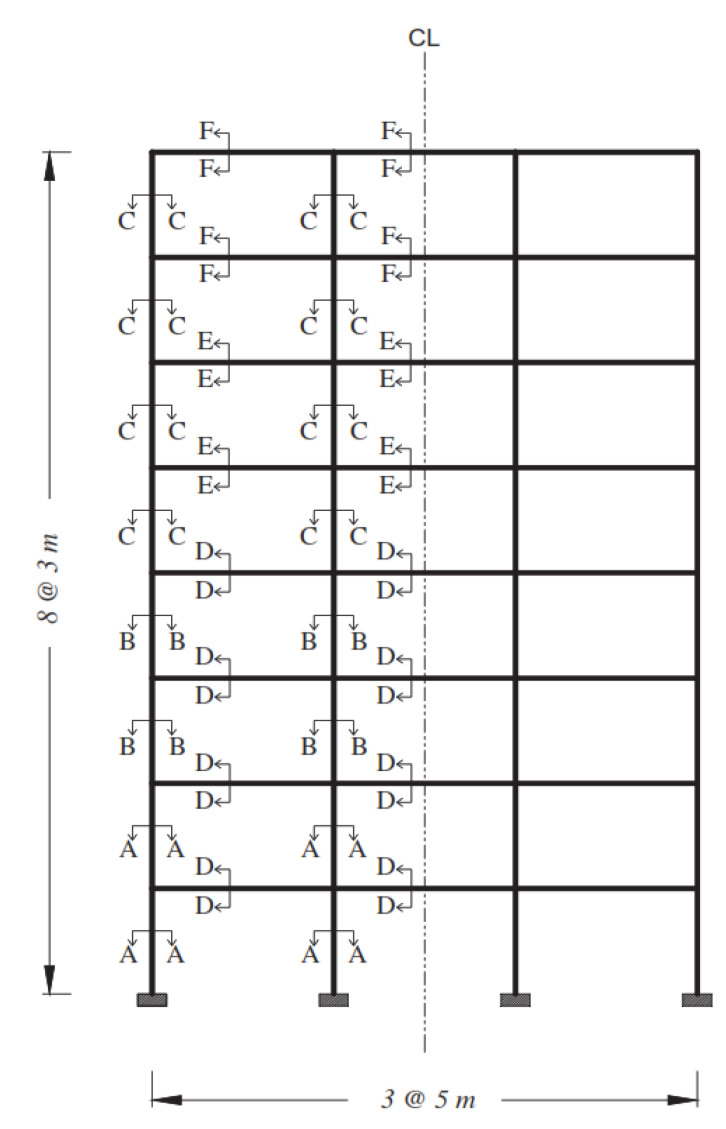
The selected moment-resisting frame [34].

**Figure 6 polymers-15-00618-f006:**
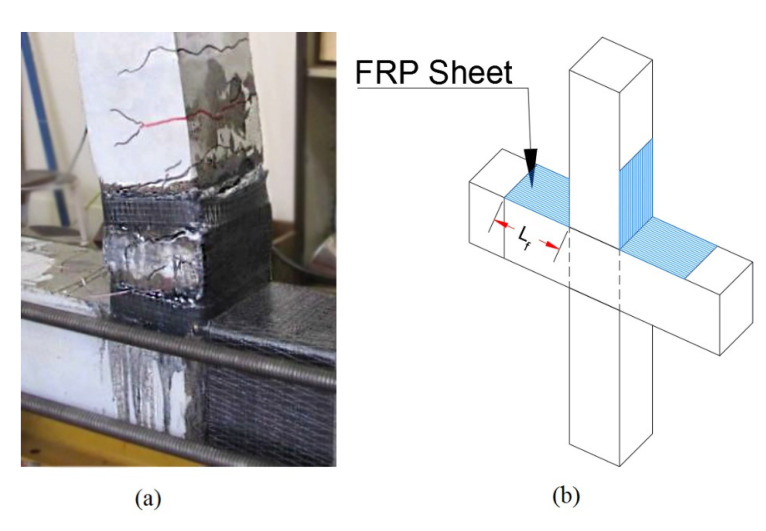
(**a**) Flange-bonding CFRP and GFRP retrofitting system [34]. (**b**) Schematic scheme of flange-bonding FRP retrofit in an interior joint.

**Figure 7 polymers-15-00618-f007:**
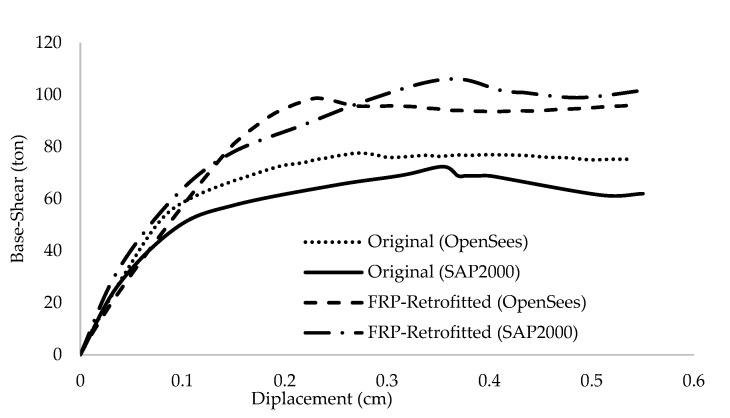
The results of the pushover analysis for original and wed-bonded CFRP frames.

**Figure 8 polymers-15-00618-f008:**
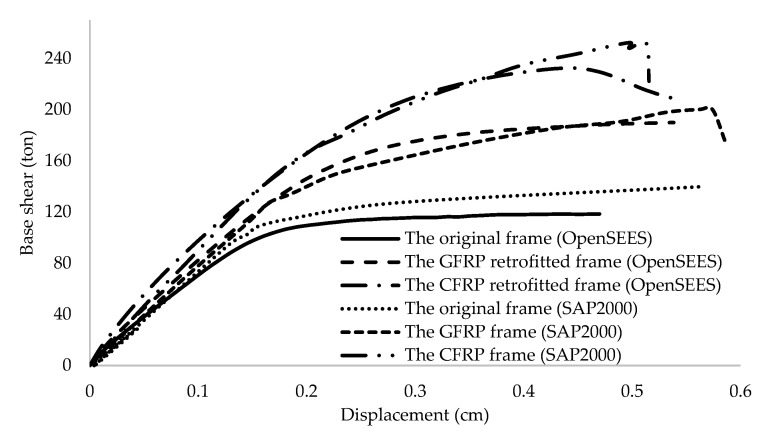
The results of the pushover analysis for original and flange-bonded CFRP and GFRP frames.

**Figure 9 polymers-15-00618-f009:**
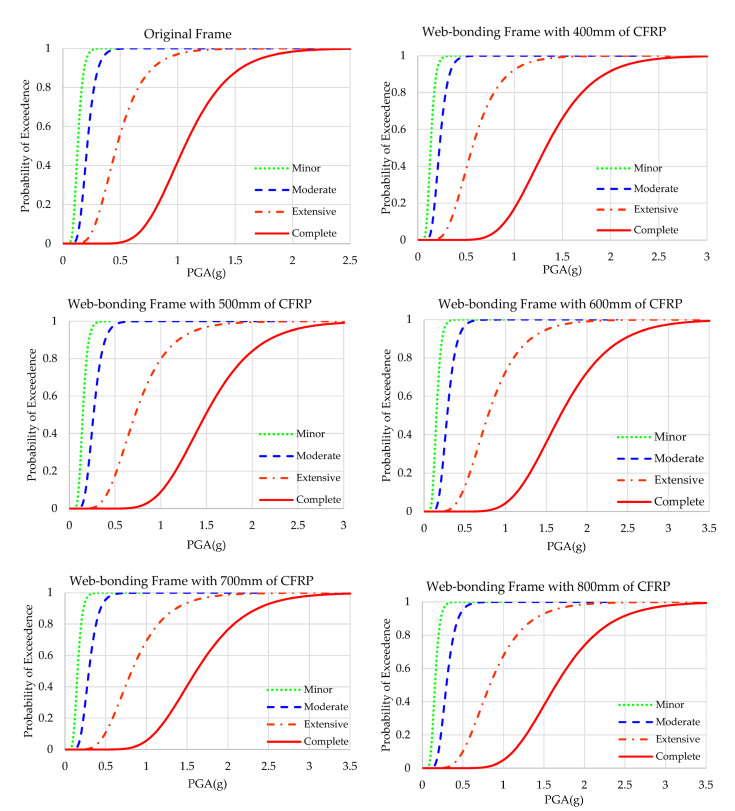
Fragility curves of the original and CFRP web-bonding frames.

**Figure 10 polymers-15-00618-f010:**
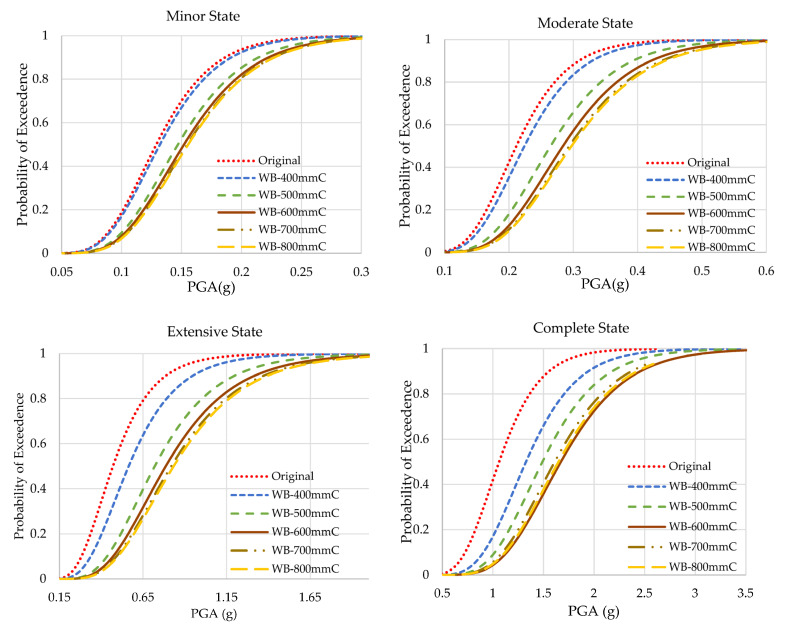
Fragility curves of the original and CFRP web-bonding frames in each state.

**Figure 11 polymers-15-00618-f011:**
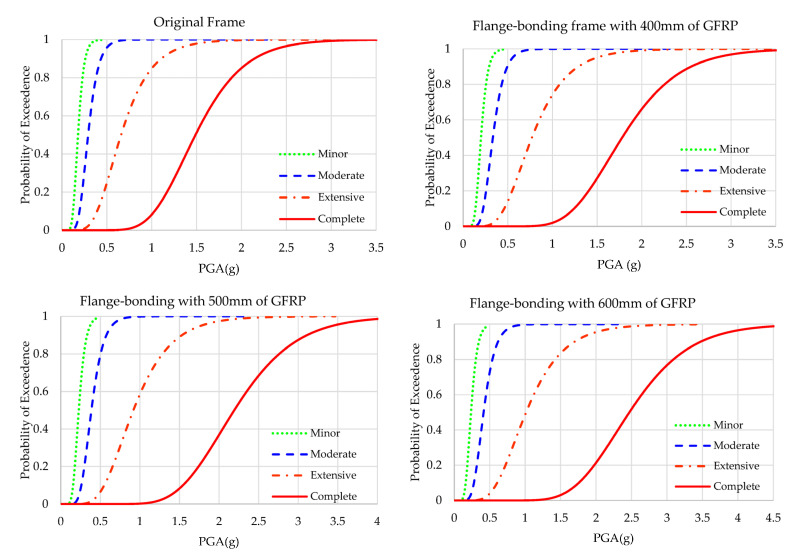
Fragility curves of the original and GFRP flange-bonding frames.

**Figure 12 polymers-15-00618-f012:**
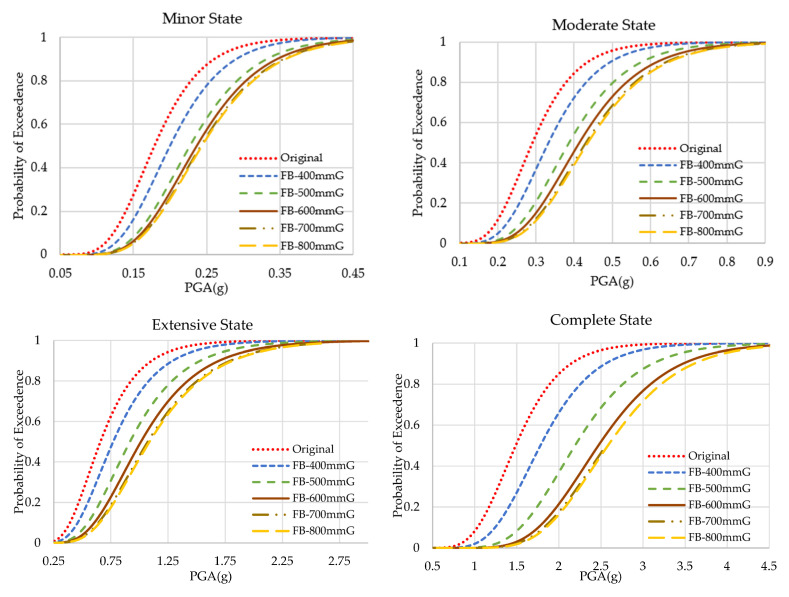
Fragility curves of the original and GFRP flange-bonding frames in each state.

**Figure 13 polymers-15-00618-f013:**
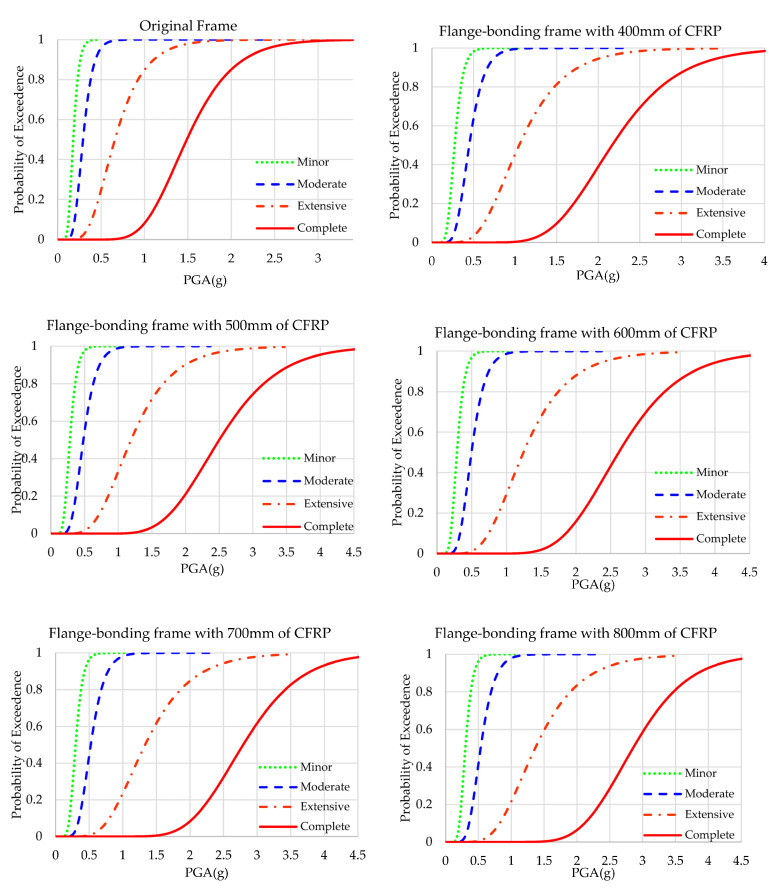
Fragility curves of the original and CFRP flange-bonding frames.

**Figure 14 polymers-15-00618-f014:**
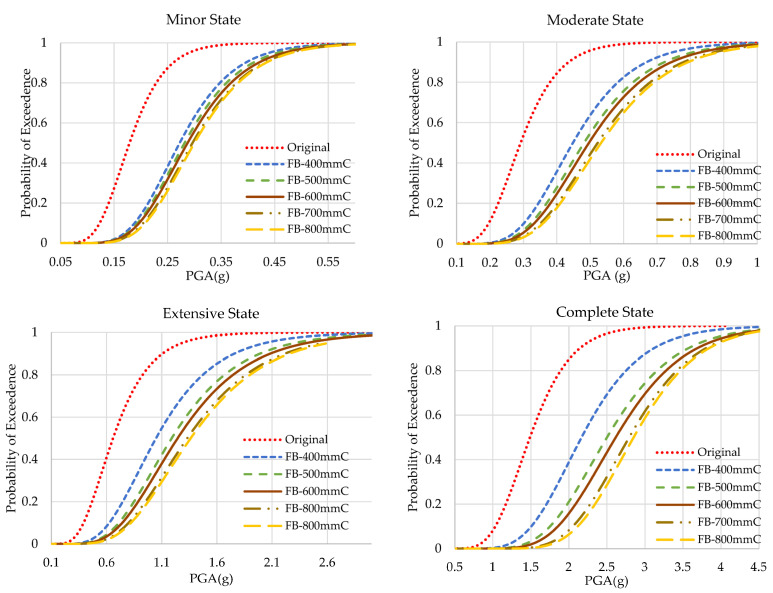
Fragility curves of the original and CFRP flange-bonding frames in each state.

**Figure 15 polymers-15-00618-f015:**
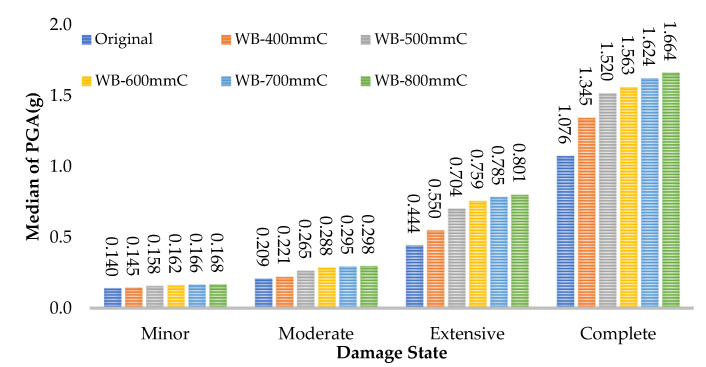
Improvement in the median of PGA in damage states in CFRP web-bonding frames.

**Figure 16 polymers-15-00618-f016:**
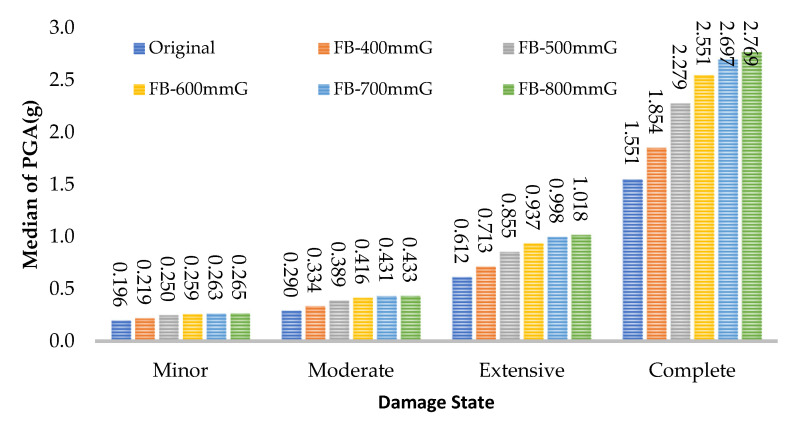
Improvement in the median of PGA in damage states in GFRP flange-bonding frames.

**Figure 17 polymers-15-00618-f017:**
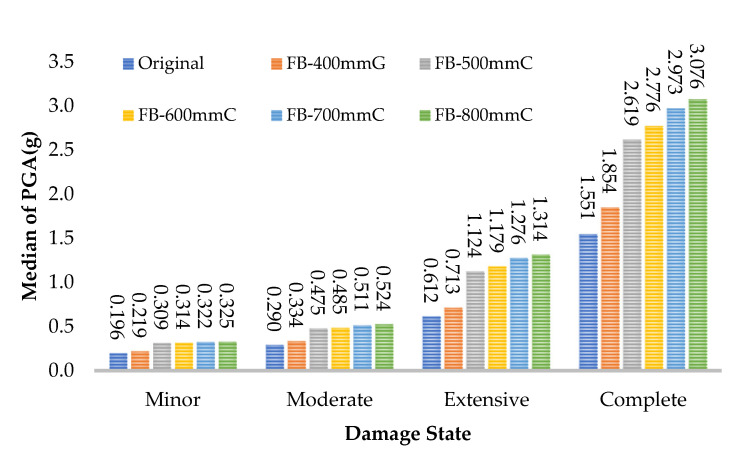
Improvement in the median of PGA in damage states in CFRP flange-bonding frames.

**Figure 18 polymers-15-00618-f018:**
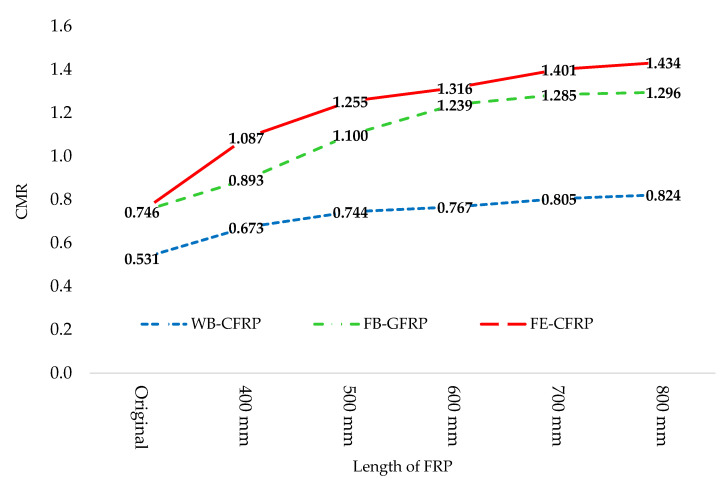
The enhancement of CMR in the retrofitted frames.

**Table 1 polymers-15-00618-t001:** Properties of the beam and column sections (mm) [7,28].

Section	b	H	d	d′	A_st_	A_s_	A′_s_
A-A	700	700	650	50	0.019	-	-
B-B	500	500	450	50	0.022	-	-
C-C	500	500	450	50	0.013	-	-
D-D	500	500	450	50	-	0.012	0.009
E-E	500	500	450	50	-	0.011	0.0071
F-F	500	500	450	50	-	0.0071	0.004

**Table 2 polymers-15-00618-t002:** Mechanical properties of CFRP sheets used for modelling [3].

Direction	Modulus of Elasticity (MPa)	Tensile Strength (MPa)	Compressive Strength (MPa)	Shear Modulus (MPa)	Poisson’s Ratio
In fibres direction	E_x_ = 240,000	σ′_x_ = 3900	σ_x_ = 80	G_xy_ = 12,576	ν_xy_ = 0.2
Perpendicular to fibres direction	E_y_ = 18,581	σ′_y_ = 53.7	σ_y_ = 80	G_xz_ = 12,576	ν_xz_ = 0.2
	E_z_ = 18,581	σ′_z_ = 53.7	σ_z_ = 80	G_yz_ = 7147	ν_yz_ = 0.3

**Table 3 polymers-15-00618-t003:** Exterior and interior joints and the FRP sheet thicknesses at each joint [3].

Section	A-A	B-B	C-C			D-D		E-E		F-F	
Story No.	1 and 2	3 and 4	5	6	7	1 and 2	3 and 4	5	6	7	8
Thickness of CFRP layers (mm)	4.96	3.3	3.3	2.475	1.65	4.96	3.3	3.3	2.475	1.65	-

**Table 4 polymers-15-00618-t004:** The details of the designed frame (mm) [34].

Section	b	H	d	d′	A_st_	A_s_	A′_s_	Shear Steel Spacing
A-A	600	600	540	60	16Φ25	-		150
B-B	600	600	540	60	16Φ18	-		150
C-C	500	500	440	60	16Φ16	-		125
D-D	500	500	440	60	-	16Φ25	4Φ25	100
E-E	500	500	440	60	-	16Φ22	4Φ22	100
F-F	500	500	440	60	-	16Φ18	3Φ18	100

**Table 5 polymers-15-00618-t005:** Properties of the far-field record set used in IDA.

No.	Record Name	Magnitude	Year	Station	Site Class (NEHRP)	PGA_max_ (g)
**1**	Northridge	6.7	1994	Beverly Hills	D	0.52
**2**	Northridge	6.7	1994	Canyon Country	D	0.48
**3**	Duzce	7.1	1999	Bolu	D	0.82
**4**	Hector Mine	7.1	1999	Hector	C	0.34
**5**	Imperial Valley	6.5	1979	El Centro Array #11	D	0.38
**6**	Imperial Valley	6.5	1979	Delta	D	0.35
**7**	Kobe	6.9	1995	Nishi-Akashi	C	0.51
**8**	Kobe	6.9	1995	Shin-Osaka	D	0.24
**9**	Kocaeli	7.5	1999	Duzce	D	0.36
**10**	Kocaeli	7.5	1999	Arcelik	C	0.22
**11**	Landers	7.3	1992	Yermo Fire Station	D	0.24
**12**	Landers	7.3	1992	Coolwater	D	0.42
**13**	Loma Prieta	6.9	1989	Capitola	D	0.36
**14**	Loma Prieta	6.9	1989	Gilroy Array #3	D	0.56
**15**	Manjil	7.4	1990	Abbar	C	0.51
**16**	Superstation Hills	6.5	1987	El Centro Imp. Co, Cent.	D	0.36
**17**	Superstation Hills	6.5	1987	Poe Road	D	0.45
**18**	Cape Mendocino	7.0	1992	Rio dell overpass	D	0.55
**19**	Chi Chi	7.6	1999	CHY101	D	0.44
**20**	Chi Chi	7.6	1999	TCU045	C	0.51
**21**	San Fernando	6.6	1971	LA-Hollywood Stor FF	D	0.21
**22**	Friuli	6.5	1976	Tolmezzo	C	0.35

**Table 6 polymers-15-00618-t006:** Structural fragility curve parameters.

Building Properties	Interstory Drift Threshold of Damage State			
Type	Minor	Moderate	Extensive	Complete
C1H	0.0020	0.0043	0.0117	0.0300

**Table 7 polymers-15-00618-t007:** The PGA (g) accordance to the probability of 50% and 100% of each state for WB-CFRP frames.

Frame	Minor State		Moderate State		Extensive State		Complete State	
	50%	100%	50%	100%	50%	100%	50%	100%
Original	0.132	0.265	0.215	0.460	0.472	1.258	1.060	2.425
WB-400mmc	0.134	0.266	0.225	0.480	0.556	1.557	1.319	2.831
WB-500mmc	0.153	0.283	0.265	0.550	0.710	1.837	1.494	3.188
WB-600mmc	0.155	0.290	0.295	0.575	0.780	2.007	1.613	3.307
WB-700mmc	0.156	0.292	0.298	0.575	0.843	2.012	1.648	3.489
WB-800mmc	0.156	0.292	0.300	0.578	0.845	2.012	1.655	3.489

**Table 8 polymers-15-00618-t008:** The PGA (g)’s accordance with the probability of 50% and 100% of each state for FB-GFRP frames.

Frame	Minor State		Moderate State		Extensive State		Complete State	
	50%	100%	50%	100%	50%	100%	50%	100%
Original	0.179	0.389	0.295	0.660	0.661	1.781	1.501	3.069
FB-400mmG	0.197	0.409	0.335	0.710	0.762	2.257	1.780	3.556
FB-500mmG	0.231	0.450	0.385	0.805	0.892	2.516	2.194	4.159
FB-600mmG	0.243	>0.450	0.434	0.825	0.990	2.745	2.467	>4.500
FB-700mmG	0.245	>0.450	0.435	0.875	1.075	2.747	2.593	>4.500
FB-800mmG	0.246	>0.450	0.436	0.900	1.077	2.747	2.593	>4.500

**Table 9 polymers-15-00618-t009:** The PGA (g)’s accordance with the probability of 50% and 100% of each state for FB-CFRP frames.

Frame	Minor State		Moderate State		Extensive State		Complete State	
	50%	100%	50%	100%	50%	100%	50%	100%
Original	0.179	0.389	0.295	0.660	0.661	1.781	1.501	3.069
FB-400mmC	0.276	0.548	0.445	0.925	1.053	2.695	2.173	4.350
FB-500mmC	0.287	0.598	0.448	0.977	1.232	2.726	2.446	>4.500
FB-600mmC	0.293	0.600	0.510	>1.000	1.235	2.779	2.607	>4.500
FB-700mmC	0.295	0.600	0.525	>1.000	1.341	>3.000	2.845	>4.500
FB-800mmC	0.296	0.600	0.530	>1.000	1.347	>3.000	2.852	>4.500

**Table 10 polymers-15-00618-t010:** Number of plastic hinges in the web and flange-bonding frames.

Type	Frames Retrofitted with Different Lengths	Number of Plastic Hinges in Columns					Total Number of Plastic Hinges in Columns	Number of Plastic Hinges in Beams					Total Number of Plastic Hinges in Beams	Total Number
		B	IO	LS	D	E		B	IO	LS	D	E		
Web-bonding frame	Original	11	8	2	0	0	21	21	6	1	8	8	44	63
	400 mm	19	2	2	0	0	23	12	16	6	9	0	43	66
	500 mm	20	2	2	0	0	24	21	18	3	5	0	47	71
	600 mm	19	4	2	0	0	25	24	13	3	3	0	43	68
	700 mm	17	4	2	0	0	23	22	11	4	2	0	39	62
	800 mm	17	4	2	0	0	23	25	9	3	2	0	39	62
Flange-bonding GFRP frames	Original	11	9	0	0	4	24	10	19	6	0	3	38	62
	400 mm	13	14	1	2	1	31	15	21	7	2	1	46	77
	500 mm	14	15	1	1	1	32	15	20	7	2	0	44	76
	600 mm	14	14	2	1	0	31	18	22	4	0	0	44	75
	700 mm	15	15	2	0	0	32	20	22	3	0	0	45	77
	800 mm	15	14	2	0	0	31	21	20	3	0	0	44	75
Flange-bonding CFRP frames	Original	11	9	0	0	4	24	10	19	6	0	3	38	62
	400 mm	13	11	1	1	1	27	16	21	5	2	1	45	72
	500 mm	14	11	1	1	0	27	16	22	7	1	0	46	73
	600 mm	15	9	1	1	0	26	16	22	6	1	0	45	71
	700 mm	14	9	2	0	0	25	17	21	6	0	0	44	69
	800 mm	15	8	2	0	0	25	19	20	5	0	0	44	69

**Table 11 polymers-15-00618-t011:** Maximum effective FRP strain in web-bonding frames.

FRP Sheet Thickness (mm)	Number of Layer	Effective Strain in the FRP Sheet		Maximum Strain (0.9 *ε_fu_*) Based on ACI 440.2-2017
		Equation (5)	OpenSees	
1.65	1	0.0033	0.0038	0.015
2.475	1	0.0027	0.0031	0.015
3.3	1	0.0023	0.0025	0.015
4.96	1	0.0019	0.0022	0.015

**Table 12 polymers-15-00618-t012:** Maximum effective FRP strain in Flange-bonding frames.

FRP Sheet Thickness (mm)	Number of Layer	Effective Strain in the FRP Sheet		Maximum Strain (0.9 *ε_fu_*) Based on ACI 440.2-2017
		Equation (5)	OpenSees	
0.165	1	0.0072	0.0085	0.014
0.165	1	0.0052	0.0063	0.014
0.589	6	0.0040	0.005	0.04
0.589	9	0.0033	0.0041	0.04

## Data Availability

Not applicable.

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
