# Peer review of "Enhancement of the Fragility Capacity of RC Frames Using FRPs with Different Configurations at Joints"

_polymers, 2023, doi:10.3390/polym15030618_

Round 1
Reviewer 1 Report
The fragility capacity of RC frames using FRPs with different configurations at joints were investigated through the numerical simulation method. Some research results obtained are important to reveal the effect of FRP configurations on the seismic performance and fragility of concrete members. The authors are suggested to consider the following comments for further supplement and improvement.
1. Abstract, this paper mainly uses the simulation method to study the fragility of FRP strengthening of concrete members. The authors were advised to further provide the accuracy of the model by comparing it with the experimental results. In addition, the performance improvement mechanism of concrete members after the strengthening should be analyzed and summarized in detail.
2. The keywords should avoid the duplication, and should accurately reflect the research work. It is recommended that further adjustments and modifications should be made.
3. Introduction
(1) The first occurrence of a noun should be given its full name, such as RC and FRP, etc.
(2) The first paragraph is too long to describe the relevant research background. It is recommended to divide it into two or more paragraphs.
(3) When summarizing the properties and advantages of FRP, it is recommended to describe it separately according to the type of fiber, such as CFRP and GFRP related to this paper. This is because the performance, advantages and major application fields of CFRP and GFRP are different due to their different performance, cost and composition. For example, CFRP has higher mechanical properties, excellent fatigue resistance, corrosion resistance and creep resistance. However, the high price and low elongation at break of carbon fiber is a major drawback. In contrast, GFRP has low price and high elongation at break, but its performance is not as good as that of CFRP. Please review the following research to supplement this part. Composite Structures, 2022, 293, 115719. https://doi.org/10.1155/2020/3495276. Polymer testing, 2020; 90: 106761.
(4) In the second and third paragraphs, after analyzing the research from the others’ work, the authors should make a summary and propose the unsolved problems to further emphasize the significance and importance of the current research work. At present, the summary is difficult for readers to better understand the unresolved problems and the importance of this study.
4. Please improve the quality of figure 2, 3 and check the full text at the same time. Other unclear pictures should be replaced if necessary. In addition, please add the units of data into the figures in the tables, such as Table 1.
5. In part 2.1, CFRP is adopted as the retrofitting materials (Table 2). In part 2.2, CFRP and GFRP are adopted as the retrofitting materials (Line 137-138 and figure 6). In addition, the numerical simulation also mainly uses CFRP and GFRP as the retrofitting materials, so how to match this different? Do you want to study the effects of different FRPs on the performance of reinforced concrete members? It is recommended to provide relevant explanations.
6. In part 3.1.1, please provide the basic performance parameters of materials. In addition, some details and method in the modeling process are further provided a reference for readers.
7. In figure 7 and 8, the authors claim “the results are close which shows that there is a good agreement between the outcomes of pushover analysis and also indicates that the OpenSees modeling is reliable”, please further provide the quantitative agreement degree between the two.
8. In part 4.1, the effects of seismic magnitude and reinforcement size of CFRP on the fragility curves have been analyzed as shown in figure 9 and 10. It is suggested to provide a table to quantitatively analyze the effect of these parameters. This is because it is difficult to understand the effect of parameters only through the comparison of the curves. Similar recommendations also apply to Section 4.2.
9. Conclusions should be condensed only include 3~4 key points according to the important findings of this paper.
Author Response
Response to Reviewer 1 Comments
Point 1: Abstract, this paper mainly uses the simulation method to study the fragility of FRP strengthening of concrete members. The authors were advised to further provide the accuracy of the model by comparing it with the experimental results. In addition, the performance improvement mechanism of concrete members after the strengthening should be analyzed and summarized in detail.
Response 1: Many thanks for your valuable comment. In this study, several 8-stoery RC buildings have been investigated through FE modelling. There is a very limited research and experimental investigations into the behaviour of a full scale FRP retrofitted frames in the literature. However, in this study, the FE models have been validated using data taken from the following sources:
- Niroomandi, A.; Maheri, A.; Maheri, M.R.; Mahini, S.S. Seismic performance of ordinary RC frames retrofitted at joints by FRP sheets. Engineering Structures 2010, 32, 2356-36.
- Ronagh, H.R.; Eslami, A. Flexural retrofitting of RC buildings using GFRP/CFRP–A comparative study. Composites Part B: Engineering 2013, 46, 188-96.
In addition, as advised, three tables have been added in the manuscript; in Sections 4.1 and 4.2; to compare the fragility performance of the FRP-retrofitted frames.
Point 2: The keywords should avoid the duplication, and should accurately reflect the research work. It is recommended that further adjustments and modifications should be made.
Response 2: Thanks for your recommendation. Keywords have been revised.
Point 3 (1): The first occurrence of a noun should be given its full name, such as RC and FRP, etc.
Response 3(1): Thanks for your recommendation. The first occurrence of all nouns, have been revised in the manuscript.
Point 3 (2): The first paragraph is too long to describe the relevant research background. It is recommended to divide it into two or more paragraphs.
Response 3(2): Thanks for your advice. We devided it into two paragraphs.
Point 3 (3): When summarizing the properties and advantages of FRP, it is recommended to describe it separately according to the type of fiber, such as CFRP and GFRP related to this paper. This is because the performance, advantages and major application fields of CFRP and GFRP are different due to their different performance, cost and composition. For example, CFRP has higher mechanical properties, excellent fatigue resistance, corrosion resistance and creep resistance. However, the high price and low elongation at break of carbon fiber is a major drawback. In contrast, GFRP has low price and high elongation at break, but its performance is not as good as that of CFRP. Please review the following research to supplement this part. Composite Structures, 2022, 293, 115719. https://doi.org /10.1155/2020/3495276. Polymer testing, 2020; 90: 106761.
Response 3(3): Thanks for the comment. An explanation on this issue, has been added to Section 1. Introduction; of the manuscrpit.
Point 3(4): In the second and third paragraphs, after analyzing the research from the others’ work, the authors should make a summary and propose the unsolved problems to further emphasize the significance and importance of the current research work. At present, the summary is difficult for readers to better understand the unresolved problems and the importance of this study.
Response 3(4): As requested, a summary has been added to the introduction.
Point 4: Please improve the quality of figure 2, 3 and check the full text at the same time. Other unclear pictures should be replaced if necessary. In addition, please add the units of data into the figures in the tables, such as Table 1.
Response 4: The units have been added to all tables and figures where required. The quiality of Figures have been improved.
Point 5: In part 2.1, CFRP is adopted as the retrofitting materials (Table 2). In part 2.2, CFRP and GFRP are adopted as the retrofitting materials (Line 137-138 and figure 6). In addition, the numerical simulation also mainly uses CFRP and GFRP as the retrofitting materials, so how to match this different? Do you want to study the effects of different FRPs on the performance of reinforced concrete members? It is recommended to provide relevant explanations.
Response 5: The main objective of this study is to investigate the effect of length of FRP sheets on the fragility capacity of RC frames and the improvement of this parameter. However, for Flange-bonding frames, a comparison between CFRP and GRFP-retrofitted RC frames, have been also carried out to study the effect of different materials on the fragility capacity of RC frames. An explanation is added to Section 3.2 of the manuscript.
Point 6: In part 3.1.1, please provide the basic performance parameters of materials. In addition, some details and method in the modeling process are further provided a reference for readers.
Response 6: As requested, this section has been revised to include more information and details of the materials.
Point 7: In figure 7 and 8, the authors claim “the results are close which shows that there is a good agreement between the outcomes of pushover analysis and also indicates that the OpenSees modeling is reliable”, please further provide the quantitative agreement degree between the two.
Response 7: Thanks for your recommendation. In Section 3.2, quantitative parameters have been added to the manuscript to demonstrate better agreements of the results in Figures 7 and 8.
Point 8: In part 4.1, the effects of seismic magnitude and reinforcement size of CFRP on the fragility curves have been analyzed as shown in figure 9 and 10. It is suggested to provide a table to quantitatively analyze the effect of these parameters. This is because it is difficult to understand the effect of parameters only through the comparison of the curves. Similar recommendations also apply to Section 4.2.
Response 8: As advised, three additional tables showing the probability of 50 and 100% of occurrence of complete collapse with explanation, have been added to Sections 4.1 and 4.2.
Point 9: Conclusions should be condensed only include 3~4 key points according to the important findings of this paper.
Response 9: The conclusion has been revised as advised.
Please note that all changes have been highlighted in yellow in the manuscript.

Reviewer 2 Report
Manuscript number: polymers-2121881
(Enhancement of the fragility capacity of RC frames using FRPs 2 with different configurations at joints)
The paper illustrates the results of an investigation into the effectiveness of different lengths of Fiber Reinforced Polymer sheets in retrofitting the joints to improve the fragility function of ordinary RC frames.
The paper is interesting, however before considering for considering it suitable for publication in the journal some suggestions should be incorporated in the manuscript.
Remarks:
1. English should and some typo mistakes should be removed.
2. What is main novelty in present work?
3. Discuss possible reason of variation of results in Figure 7 and 8.
4. Can this study results extended for 25 storey building also?Discuss.
5. Write conclusions pointwise.
6. Some recent works should be included in manuscript.
Investigation of porosity effect on flexural analysis of doubly curved FGM conoids. Science and Engineering of Composite Materials 2019, 26, 435–448.
Flexural analysis of functionally graded CNT reinforced doubly curved singly ruled composite truncated cone. Journal of Aerospace Engineering 2019, 32, 040181541-0401815411.
Author Response
Response to Reviewer 2 Comments
Point 1: English should and some typo mistakes should be removed.
Response 1: The manuscript has been fully revised for English proficiency.
Point 2: What is main novelty in present work?
Response 2: In this study, the effect of different length of FRP sheets on the fragility capacity of a RC frame, have been investigated. The effective length is an important parameter on the fragility behavior of RC frames which has less been investigated previously.
In this regards, a statement, has been added to the Introduction of the manuscript to address this novelty.
Point 3: Discuss possible reason of variation of results in Figure 7 and 8.
Response 3: In this study, several FRP-retrofitted frames, have been modeled. The quantitative comparison has been added to the manuscript to compare the results more accurately. One possible reason for the variation, is that there is no direct method to model FRP materials in SAP2000, whereas, this is possible with OpenSees.
Point 4: Can this study results extended for 25 storey building also?Discuss.
Response 4: Practically, running such an analysis for a 25-storey frame is very difficult and is a time-consuming process. Modelling and running models for this study takes nearly 6 months with a powerful PC. Therefore, it might not be possible or easy to carry out this analysis for 25-storey buildings. However, the results can be used for RC frames with 25 stories with caution.
Point 5: Write conclusions pointwise.
Response 5: Many thanks for your recommendation. The Conclusion has been revised as noted.
Point 6 : Some recent works should be included in manuscript.
Response 6: As requested, several up-to-date studies, have been added to the manuscript..
Please note all changes have been highlighted in yellow in the manuscript.

Reviewer 3 Report
1: The English should enhanced as well.
2. There is no research on the enhancement mechanism in this paper, and the author needs to analyze the enhancement mechanism more carefully.
3. The author lacks logicality in the introduction of the foreword, so the author needs to quote papers in this field to enhance logicality.
4. Please clearly introduce the strengthening mechanism of FRPs 2.
Author Response
Response to Reviewer 3 Comments
Point 1: The English should enhanced as well.
Response 1: Many thanks for your advice. The manuscript has been fully revised for English proficiency.
Point 2: here is no research on the enhancement mechanism in this paper, and the author needs to analyze the enhancement mechanism more carefully.?
Response 2: As advised, in order to show the enhancement mechanism, three tables showing the probability of 50 and 100% of occurrence of complete collapse, have been added to Sections 4.1 and 4.2 with explanation.
Point 3: The author lacks logicality in the introduction of the foreword, so the author needs to quote papers in this field to enhance logicality.
Response 3: Some modifications have been made in the Introduction section and some relevant studies have been added with more explanations regarding the importance of this study and its novelty.
Point 4: Please clearly introduce the strengthening mechanism of FRPs 2.
Response 4: Many thanks for your advice. An explanation regarding strengthening mechanisms have been added to Sections 4.1 and 4.2 .
Please note that all changes have been highlighted in yellow in them anuscript.

Round 2
Reviewer 1 Report
The authors have provided a detailed reply based on most of the comments. The following minor comments still need further improvement.
Point 1: As the authors reply, the FE models have been further verified by the experimental results in the literature. It is suggested that the authors reflect this fact in the abstract to make the accuracy of finite element simulation more convincing.
Point 3(3), the performance, advantages and the main application fields of CFRP and GFRP have been summarized by the authors. The above statement and summary should be verified by the research conclusions from the literature. In addition, the disadvantages of CFRP and GFRP should be objectively reflected, such as the higher price and lower elongation at break of CFRP, the relatively poor corrosion resistance of GFRP, etc.
Point 4, the quality of Figures 1 and 2 are still unclear after being slightly enlarged, this may affect the review effect after the publications.
Point 5, For the RC frames, the effect of length of FRP sheets on the fragility capacity was investigated. For flange-bonding frames, the effect of different materials on the fragility capacity was investigated. If the length and type of FRP sheets are considered at the same time, what is the effect on the friability? In addition, the reviewer did not see the explanation was added to Section 3.2 of the manuscript.
Author Response
Response to Reviewer 1-2 Comments
Point 1: As the authors reply, the FE models have been further verified by the experimental results in the literature. It is suggested that the authors reflect this fact in the abstract to make the accuracy of finite element simulation more convincing.
Response 1: Thanks for your valuable comment. An explanation has been added to the abstract.
Point 3(3): The performance, advantages, and main application fields of CFRP and GFRP have been summarized by the authors. The above statement and summary should be verified by the research conclusions from the literature. In addition, the disadvantages of CFRP and GFRP should be objectively reflected, such as the higher price and lower elongation at the break of CFRP, the relatively poor corrosion resistance of GFRP, etc.
Response 3(3): Thanks for your recommendation. The performance of GFRP and CFRP has been verified by the research conclusions from the literature as requested. Please see lines 43-48 of the revised manuscript.
Point 4: The quality of Figures 1 and 2 are still unclear after being slightly enlarged, this may affect the review effect after the publications.
Response 4: As recommended, these figures have been replaced with higher-quality ones.
Point 5: For the RC frames, the effect of the length of FRP sheets on the fragility capacity was investigated. For flange-bonding frames, the effect of different materials on the fragility capacity was investigated. If the length and type of FRP sheets are considered at the same time, what is the effect on the friability? In addition, the reviewer did not see the explanation added to Section 3.2 of the manuscript.
Response 5: Many thanks for your comments. In this study, three veriable including the configuration (flange and web bonded frames) , the length (from 400 to 800mm), and the type of FRP materials (CFRP and GFRP) have been investigated. The performance and fragility curves of each category have been then analysed and determined. It was shown that which material has a better performance and for each, the impact of length has been determined and discussed. The generated curves and diagrams have provided the possibility to compare any category and model in this study.
Moreover, the added explanation to Section 3.2 has been highlighted in green on Page 9.
Please note that all changes have been highlighted in green.

Reviewer 2 Report
OK
Author Response
Many thanks for your valuable time.
Best regards,
Saeed
Round 3
Reviewer 1 Report
The authors have made necessary improvements and recommend accepting the current paper.